# Incidence of Vaccine-Preventable Childhood Diseases in the European Union and in the European Free Trade Association Countries

**DOI:** 10.3390/vaccines9070796

**Published:** 2021-07-17

**Authors:** Estera Jachowicz, Magdalena Gębicka, Daria Plakhtyr, Myroslav Shynkarenko, Juri Urbanowicz, Maciej Mach, Jacek Czepiel, Jakub Marchewka, Jadwiga Wójkowska-Mach

**Affiliations:** 1Department of Microbiology, Faculty of Medicine, Jagiellonian University Medical College, 31-121 Krakow, Poland; estera.jachowicz@doctoral.uj.edu.pl; 2Department of Immunology, Faculty of Medicine, Jagiellonian University Medical College, 31-121 Krakow, Poland; magdalena.gebicka@doctoral.uj.edu.pl; 3Faculty of Medicine, Jagiellonian University Medical College, 31-121 Krakow, Poland; daria.plakhtyr@student.uj.edu.pl (D.P.); myroslav.shynkarenko@student.uj.edu.pl (M.S.); yuriy.urbanovych@student.uj.edu.pl (J.U.); 4Student of Quantitative Methods in Economics SGH Warsaw School of Economics, 02-554 Warsaw, Poland; mach-maciej@wp.pl; 5Department of Infectious and Tropical Diseases, Jagiellonian University Medical College, 30-688 Krakow, Poland; jacek.czepiel@uj.edu.pl; 6Department of Physiotherapy, University of Physical Education, 31-571 Kracow, Poland; jakub.marchewka@awf.krakow.pl

**Keywords:** vaccine-preventable childhood diseases, incidence rate, vaccine schedules, mumps, rubella, measles, *Streptococcus pneumoniae*, *Neisseria meningitidis*

## Abstract

Introduction: Despite the widespread availability of vaccines, the incidence of vaccine-preventable childhood diseases (VPCD) started to grow in recent years. The aim of the study was to compare the annual incidence of selected VPCDs in the EU (European Union) and EFTA (European Free Trade Association) countries in the period of the last 5 years (2014–2019 or other intervals, depending on data availability), and the country-specific vaccine schedules. Methods: VPCD incidence rates in Europe were based on “The Surveillance Atlas of Infectious Diseases” by the ECDC (European Centre for Disease Prevention and Control); vaccination schedules were based on ECDC reports. Results: The obligation to vaccinate was not universal, and it generally only applied to two preparations: the MMR (measles, mumps, rubella) vaccine and the one against polio. During the study, the situation associated with mumps did not change or improve in individual countries; the median incidence amounted to 30 cases. The median incidence associated with rubella amounted to 1 case, but in a few countries, it grew very rapidly, i.e., in Germany, Italy, and Romania; in Poland, the incidence was clearly decreasing, from 5923 to 1532 cases. The most dynamic situation concerned measles. The total median was 2.4 cases per 100,000 population; the only one country with falling incidence was Germany. The diseases associated with *Streptococcus pneumoniae* and *Neisseria meningitidis* remained at a stable level in all analyzed countries. Conclusion: Vaccine schedules differ among the countries, so does the epidemiological situation of selected diseases. Morbidity on measles was the most disturbing phenomenon: the incidence rate increased in almost 40% of all countries, regardless of the obligation to vaccinate. The increasing incidence of VPCD may be due to anti-vaccine movements, the activity of which is often caused by mistrust and spreading misinformation. In order to better prevent the increase in morbidity, standardization of vaccine schedules and documentation should be considered in the EU countries.

## 1. Introduction

The introduction of vaccines against measles, rubella, mumps, polio, chicken pox, rotavirus diarrhea, as well as meningococcal and pneumococcal infections (vaccine-preventable childhood diseases, VPCD), bore the greatest importance in reducing infant and child mortality [1]. Unfortunately, despite the availability of vaccines against childhood diseases, those diseases are still quite common, and, in recent years, more and more cases were being reported despite the fact that most of them are under strict epidemiological surveillance [2]. As early as in the beginning of the twentieth century, in 1912, the United States of America saw the introduction of an obligation to report measles infections at the national level, and an average of 6000 deaths were reported annually in the first decade of reporting [3]. In 2000, the USA was declared free of the disease, but possibly due to the intensification of migration of people from countries with poor vaccination levels against measles, the incidence of this disease has increased and, in 2019, there were 1282 confirmed cases of measles in the USA [4].

The evidence on the effectiveness of vaccination and its importance for public health is the eradication of smallpox and the almost complete eradication of polio. In the early 1950s, polio epidemics caused paralysis in over 15,000 people in the USA alone. Following the introduction of the trivalent inactivated vaccine (IPV) in 1955 and the oral vaccine (OPV) in 1963, the number of cases of polio in the USA plummeted to fewer than 100 in the 1960s and fewer than 10 in the 1970s [5].

Nevertheless, modern medicine is still struggling with VPCD, and the World Health Organization (WHO) has raised the alarm that some of them have again become a real threat to the health and life of newborns and children in all WHO regions [6]. The objective of the study was to compare the incidence of selected VPCDs in the European Union (EU) and European Free Trade Association (EFTA) countries in 2014–2019, taking into account the existing vaccination programs and an attempt to evaluate whether and how the systemic surveillance of VPCDs contributes to better public health and public safety in the EU and EFTA countries.

## 2. Methodology, Data Sources

The analysis encompassed the EU (Austria, Belgium, Bulgaria, Croatia, Cyprus, Czechia (formerly called the Czech Republic), Denmark, Estonia, Finland, France, Germany, Greece, Hungary, Italy, Latvia, Lithuania, Luxembourg, Malta, the Netherlands, Poland, Portugal, Republic of Ireland, Romania, Slovakia, Slovenia, Spain, Sweden, and the United Kingdom (UK)) and EFTA countries (Switzerland, Norway, Liechtenstein, Iceland). The years selected for analysis were 2014–2019 (or shorter intervals, depending on data availability), excluding 2020 when the COVID-19 (coronavirus disease—2019) pandemic broke out, and a reduction in the vaccination coverage and decline in the total number of vaccines administered were observed worldwide [7].

For our analysis, we have chosen a few viral and bacterial acute diseases with high epidemic potential, qualifying acute diseases as developing suddenly and lasting a short time, often only a few days or weeks, and accompanied by distinct symptoms that require urgent or short-term care. The analysis covered measles, rubella, mumps, polio, chickenpox, rotavirus, and bacterial: invasive meningococcal and pneumococcal infections. Data were collected and compiled on the basis of:

VPCD Incidence Rates in Europe:Based on “The Surveillance Atlas of Infectious Diseases” by the ECDC (European Centre for Disease Prevention and Control), the information contained in the dataset provided through ATLAS is made available by ECDC collating data from the Member States collected through The European Surveillance System (TESSy) [8,9].Converted the incidence for rubella, mumps, Streptococcus pneumoniae (Belgium only) into cases per 100,000 population; the populations were estimated based on a recent ECDC report (https://www.ecdc.europa.eu/en/publications-data/data-national-14-day-notification-rate-COVID-19) (access on 30 June 2021)Long-term VPCD incidence rates (Figure 1) were obtained for:
Measles, mumps (no data for Belgium 2019), polio, chicken pox, and rotavirus diarrhea. Depending on the disease, the data for different years were available: 2015–2019 for measles, 2014–2018 for mumps, 2014–2019 for rubella, 2014–2018 for Streptococcus pneumoniae, and 2014–2017 for Neisseria meningitidis. For each disease and country, the infection cases from the available years were summed up and then presented as the number of cases in the period per 100,000 population.For rubella (no data for Croatia 2019 and Italy 2014) as well as meningococcal and pneumococcal infections, the total amounts of reported cases were summed up for each country

Due to the lack of data on the national incidence, the incidence rates of chickenpox and rotavirus were not analyzed [10]. No data were obtained as regards cases per 100,000 population for measles for 2014. Data concerning the epidemiology and incidence rates of the studied VPCDs in Switzerland were not obtained.

Vaccination Schedules for Measles, Mumps, Rubella Polio, as Well as Meningococcal and Pneumococcal Infections:Based on shared databases, ECDC reports [11].Vaccine schedule in Switzerland [12].

To compare the mandatory vs. non-mandatory policy to the annual incidence observed (cases per 100,000 population/year) we used a generalized linear regression model. Calculations were performed using Statistica 13.3 (TIBCO Software Inc., Palo Alto, CA, USA). *p* < 0.05 was considered statistically significant.

## 3. Results

### 3.1. The Incidence of VPCD

The incidence associated with the discussed infectious diseases varied greatly with no evident pattern or clear trend. Among the countries under study, only in Luxembourg and in Malta, the incidence rates observed for each of the diseases under study were low. In Austria, Belgium, Croatia, Cyprus, Denmark, Estonia, Hungary, Latvia, the Netherlands, Norway, Portugal, Slovakia, Slovenia, and Sweden, the situation was stable. In France, Italy, Germany (excluding measles), Romania, and the UK (excluding rubella), high incidence persisted (Table 1 and Table 2).

The most dynamic situation concerned measles. The median was 2.4 cases per 100,000 population, and the only one country with falling incidence was Germany: in 2015 it was 246, and there were 51 cases per 100,000 population in 2019. In half of the studied countries, increased incidence was found: in Bulgaria, Czechia, France, Slovakia, Slovenia, Spain, and the UK, where the increase was multiplied, and in Italy, Lithuania, Poland, and Romania, where the increase was bigger. The highest change in incidence was found in Italy, from 25.6 to 162 cases per 100,000 population, and in Bulgaria, with 0 and 111.9 cases per 100,000 population (Table 1). The highest incidence was observed in 2017 in Italy, 539.9, which was five times the average for that year, but the 5-year incidence rate was the highest in Romania (Figure 1).

During the study, the situation associated with mumps did not change or improve in individual countries. High incidence rates were constantly observed in Belgium, Czechia, Finland, Germany, Italy, Poland, and the Republic of Ireland. There has been a decline in Slovakia (from 28.56 in 2014 to 0.24 cases in 2018 per 100,000 population), and growth in Spain (from 2.03 in 2014 to 11.46 per 100,000 population) (Table 1). The 5-year incidence rates point to a significant problem in surveillance in Czechia and the Republic of Ireland (Figure 1).

The median incidence associated with rubella amounted to below 1 case per 100,000 population, but in a few countries, it grew rapidly, i.e., in Italy and Romania. In Poland, the incidence was clearly decreasing, from 15.6 to 4.0 cases per 100,000 population (Table 1), but the long-term incidence rate of rubella in Poland was several times higher than in other countries.

According to ECDC data, Europe has remained polio-free since 2002 [13].

The diseases associated with *Streptococcus pneumoniae* and *Neisseria meningitidis* remained at a stable level in all the studied countries; however, this level was different. The highest was in Belgium, Denmark, Finland, Netherland, and Sweden (*S. pneumoniae*) and in Lithuania (*N. meningitidis*) (Table 2). The 5-year incidence rates point to a significant problem in surveillance of *S. pneumoniae* in Slovenia and Northern Europe with Nordic countries (Denmark, Finland, Norway, and Sweden, without Iceland) and the Netherlands (Figure 1). According to *N. meningitidis* invasive diseases, the long-term incidence rate was the highest in Lithuania and the Republic of Ireland.

We found no simple correlation between mandatory vs. non-mandatory policy and the observed incidence of measles, mumps, rubella, *S. pneumoniae,* and *N. meningitides* infections in studied countries.

### 3.2. Vaccination Schedules

The obligation to vaccinate was not universal, and it generally only applied to two preparations (the MMR (measles, mumps, rubella vaccine) and the one against polio) and only to a limited extent. Countries with the highest numbers of mandatory vaccinations were in Bulgaria, Croatia, France, Hungary, Latvia, Poland, and Slovakia, where all of the discussed vaccines, except for the vaccinations against *N. meningitidis,* Rotavirus and Varicella-Zoster Virus, were mandatory, while in 19 of the countries under study there was no compulsion in this regard. In Latvia, all vaccinations were mandatory except for *N. meningitidis* (Appendix A, Appendix A).

The most common vaccination was the one against polio, which was mandatory in 11 out of 31 countries: Belgium, Bulgaria, Croatia, Czechia, France, Hungary, Italy, Poland, Slovakia, Slovenia, and Malta. The MMR vaccine, against measles, mumps, and rubella, was mandatory in 9 out of 31 countries, both inside and outside the EU, namely, in Bulgaria, Croatia, Czechia, France, Hungary, Italy, Poland, Slovakia, and Slovenia (Appendix A).

The pneumococcal vaccination was mandatory in 7 out of 31 countries (22.6%; Bulgaria, Croatia, Latvia, France, Hungary, Poland, and Slovakia). Meanwhile, the meningococcal vaccination was only imposed in France, and all 31 countries recommended these vaccinations (Appendix A).

The MMR vaccination, against measles, mumps, and rubella, was administered in all 31 countries and in France; additionally, monovalent vaccines against measles, mumps, and rubella were used in 6-month-old children in specific circumstances. Vaccination with two doses of MMR was mandatory in all countries, and additionally, in Germany, Czechia, and France, it was possible to re-vaccinate a child at a later age if a dose had been omitted before. In 18 out of 31 countries, the first dose is given at 12 months (12 countries: Belgium, Croatia, Estonia, France, Greece, Latvia, Luxembourg, Portugal, Republic of Ireland, Romania, Spain, and the UK, or 13 months (six countries: Bulgaria, Cyprus, Czechia, Italy, Malta, and Poland). The earliest, one dose was routinely administered in Austria and Switzerland (9 months), in Denmark in risk groups (9 months), and in France in specific circumstances (6 months), while it was the latest in Sweden and Iceland (18 months) (Appendix A).

As for the polio vaccine, the first dose was given to babies at 2 or 3 months of age, and only in Poland was it given only at the age of 4 months. The number of doses in different countries varies greatly. Currently, in two countries (Slovenia and Switzerland), three doses were administered. In 13 countries (Austria, Belgium, Czechia, Denmark, Finland, Greece, Spain, Republic of Ireland, Iceland, Malta, Poland, Romania, and Sweden), four doses were administered, and Malta allows an additional, optional fifth dose. In 12 countries (Bulgaria, Cyprus, Estonia, the UK, the Netherlands, Lithuania, Germany, Norway, Portugal, Slovakia, Hungary, and Italy), five doses were applied, and Germany grants a sixth, non-mandatory dose. In three countries, there were six doses. In France, babies were given three doses, followed by booster vaccinations every 5 years. Inactivated polio vaccines (IPV) are employed in all countries studied and, in the UK, attenuated live vaccines (OPV) were additionally used. Currently, in Europe, the vaccines employed were designed to work solely against polio, and there were also combined vaccines against polio and other diseases (Appendix A).

Pneumococcal vaccines were administered in all countries except for Estonia. The first dose was given at 2 or 3 months, whereas in Romania, the first dose was given only after the child turns 1 year of age. The majority of countries provided three doses, while Cyprus and the UK administer two doses (Appendix A).

Meningococcal vaccination was carried out in 19 out of 31 countries (61.3%); however, they were mandatory only in France. Depending on the type of vaccine administered (against various serotypes), immunization schedules in different countries varied greatly regarding the number of doses administered. The MenB (Serogroup B meningococcal vaccine) vaccine was administered in eight countries (Austria, Czechia, Italy, Lithuania, Luxembourg, Republic of Ireland, Malta, and the UK), and the number of doses in these countries ranged from one to four. MenC (Serogroup C meningococcal vaccine) was administered in 14 countries, and there were usually one or two doses. MCV4 (Meningococcal Conjugate Vaccine) was administered in four countries (Czechia, Greece, Malta, and the Netherlands), and there were usually 1–3 doses, depending on the country. In Switzerland, ACWY (meningococcal conjugate vaccine, serogroups ACWY) vaccine was administered for 2-year-olds, which constituted additional vaccination (Appendix A).

Vaccination against rotavirus was conducted in 16 out of 31 countries, but they were mandatory only in Latvia and recommended in the remaining countries. Additionally, in Spain, the vaccination was recommended for babies born prematurely, and in the Netherlands to children from the exposed groups. The vaccination was not conducted in Bulgaria, Croatia, Cyprus, Denmark, France, Hungary, Iceland, Malta, Portugal, Romania, Slovakia, or Slovenia. Currently, all the countries mentioned make use of oral live attenuated vaccines. In five countries, vaccinations in children were started at 6 weeks (Czechia, Germany, Norway, Spain, and Sweden), and in Poland between 6 weeks and 6 months. In 10 countries (Austria, Belgium, Estonia, Finland, Greece, Latvia, Lithuania, Luxembourg, Republic of Ireland, and the Netherlands), children were vaccinated at 2 months of age, while in Italy from 3 to 7 months, and in the UK, it is performed at 8 weeks of age. The number of doses ranged from one to three, and usually, there were two doses (Appendix A).

The vaccine against *Varicella zoster virus* was currently employed in 11 out of 31 countries, additionally in Poland and Czechia (for exposed groups) and in Belgium and Switzerland (only to people without immunity). In three countries (Hungary, Italy, and Latvia), the vaccine was mandatory. The number of doses administered, depending on the country, amounted to one or two (Appendix A).

## 4. Discussion

The epidemiological situation expressed by the incidence rate of selected childhood diseases is very uneven. It seems that one of the most dangerous diseases, polio, has been eradicated. The latest assessment by the European Regional Certification Commission for Poliomyelitis Eradication concluded that there was no wild poliovirus transmission or circulation of vaccine-derived poliovirus in the WHO European Region in the twenty-first century. However, Romania (EU) and countries bordering the EU, Ukraine, Bosnia, and Herzegovina, remain at high risk of a sustained polio outbreak [9].

The most disturbing situation concerns measles. Almost 40% of the countries studied report an increasing incidence of the disease, and this growth applies to countries where vaccination is mandatory, such as Bulgaria, Czechia, France, Italy, Poland, Slovakia, and Slovenia, as well as the ones where vaccinations are only recommended, such as Lithuania, Romania, Spain, and the UK. The situation also has no relation to the historical context, i.e., high incidence occurs in both countries of the former Eastern Bloc, among others, Lithuania, Poland, and Romania, as well as in the so-called *Old Union*, Spain, France, or Italy. However, it is difficult to escape from the historical context, as the vaccination obligation applies mainly to the countries of the former Eastern Bloc: Bulgaria, Croatia, Hungary, Latvia, Poland and Slovakia, and, exceptionally, France. On the other hand, in Bulgaria, Czechia, France, Italy, Romania, and the UK, vaccination coverage was very low in 2018, less than 90%, and in Poland, Slovakia, Slovenia, Lithuania, and Spain, it was only slightly better, below 95% [13]. We found that country-level vaccination schedule—mandatory vs. non-mandatory policy—was not associated with the incidence of measles, mumps, and rubella. The data indicate that mandatory vaccination is not the optimal solution. This is confirmed by the data by Vaz et al. in which not only mandatory vaccinations but also those that are an obligation subject to fines are associated with higher vaccination coverage. Additionally, a secondary, very strong predictor for incidence rate was mandatory vaccination but with nonmedical exemptions [14]. Furthermore, mandatory vaccination policy is not necessarily correlated with vaccination coverage. Multivariate models with adjustments for epidemiologic, socioeconomic factors should be performed in order to comprehensively evaluate associations between incidence, policy, and vaccination coverage.

Migration may be another reason for the wide variation regarding the measles incidence rate. For example, Italy—where high incidence rates were reported for measles, mumps, and rubella—is a country with a large number of foreign residents who arrive from outside the EU, including, among others, North Africa [15]. Unfortunately, most European countries screening migrants focus on single diseases only, mainly active or latent tuberculosis, and are particularly targeted at asylum seekers or refugees. Screening for other diseases, including VPCD, is rare [16]. The subject requires further research. Czech studies show a poor response to mumps vaccination [17,18].

The above elements, which could have an impact on the effectiveness of surveillance of the VPCD, do not explain the situation observed in Poland, where the 5-year incidence of rubella and mumps was several times higher than in the EU. It was also high in the closest EU neighbors’ countries. This might also be caused by the relatively recent introduction of the MMR vaccine for all children in Poland (2004).

Therefore, compliance with the vaccination schedules applied is perhaps important; however, vaccination schedules for measles, rubella, mumps, pneumococci, and meningococci in Europe and other countries studied are very similar, and relatively minute differences concern the schedule of administration of the preparations (ages at which subsequent doses are administered/number of doses) and reimbursement—the level of co-payment for the vaccine and its administration and the obligation or voluntary administration of the vaccine—which is especially true of pneumococcal and meningococcal vaccines.

The epidemiology of two diseases caused by encapsulated bacteria, Streptococcus pneumoniae and *Neisseria meningitidis*, was characterized by a great diversity throughout the European Union, possibly related to the high variability of their serotypes, which additionally vary in different populations. In Europe, invasive pneumococcal disease (IPD), caused by Streptococcus pneumoniae, is one of the main causes of meningitis or invasive inflammatory diseases of various organs in children and community-acquired pneumonia in adults. Currently, the number of serotypes of these bacteria is estimated at over 90, which implies enormous problems in the practical application of the available vaccines [19]. The three types of pneumococcal vaccines which are available on the market differ regarding the number of capsular serotypes combined with a carrier protein contained in the preparation. According to an ECDC report, in 2018, the most common serotypes among EU infants and children aged 1–4 included 8, 10A, 3, 19A, and 24F. Those serotypes are not included in any of the currently licensed PCVs—with the exception of 19A [20]. Serotype 24F is the most frequently isolated *S. pneumoniae* serotype in Denmark [21]. Only 17% of *S. pneumoniae* isolates in Sweden are contained in the PCV13 vaccine [22].

A high number of *S. pneumoniae* cases has been a significant problem in Slovenia for many years. National Vaccine Recommendations Program against *S. pneumoniae* was introduced in 2015, but vaccination coverage is still moderate (49–55%). Unfortunately, in addition, there is an increased number of cases caused by penicillin-resistant strains, possibly due to the overuse of antibiotics [23].

*Neisseria meningitidis*, causing invasive meningococcal disease (IMD), has 13 serotypes. Serotype A is responsible for the majority of the illnesses, and it is also accountable for cyclical epidemics of meningitis in North Africa every 5–10 years and, in the past 25 years, in China and Russia. Serotype A was also responsible for individual IMD outbreaks in Europe until the mid-twentieth century. Diseases caused by serotype B are rarer; however, its long-term outbreaks are characterized by high morbidity and mortality [24]. At the moment, most of the IMD cases in Europe are caused by serogroup B and C meningococcal strains [25]. Unfortunately, within meningococci, high genetic and antigenic variability was observed associated with gene transfer and chromosomal mutations. In recent years, the Scandinavian MenY (Serogroup Y meningococcal) and MenW (Serogroup W meningococcal) serogroups have acquired special importance. The latter was initially responsible for the meningitis outbreak among the Hajj pilgrims to Mecca in 2000, and according to Booy et al., this serotype is currently the main cause of IMD in Europe, South America, Australia, and some parts of Africa [26]. According to ECDC reports, an increased incidence of meningitidis has been observed in Lithuania since 2013. In this country, the most common serotype responsible for the disease was B, infecting mainly young children. However, vaccination against meningitidis B has only been routinely introduced in Lithuania since 2018 [27].

### 4.1. Socioeconomic Factors

It is estimated that socioeconomic factors determine the health of the population to the extent of 40% in the USA [28], and also in Poland, one’s health condition is closely related to the level of education, income, and social standing [29]. The economic status of an individual also strongly affects the epidemiology of infectious diseases in developing countries, where, for instance, wealthy people can rarely be found in crowded places and make use of public transportation less often, which significantly reduces the risk of transmission of microorganisms [30]. Furthermore, access to a doctor lowers the risk of infection, and data from Cambodia indicate a reduction in the risk of infection of even 10%. This is due to both access to a higher level of healthcare infrastructure, e.g., a properly equipped doctor’s practice or vaccination center, but also access to specific (vaccinations) or non-specific (hand hygiene and environmental hygiene education) prevention of infections [30]. It is not known whether such dependencies are observed in European countries, but for sure, access to a doctor is limited by an insufficient number of doctors in some countries, e.g., in Poland, there are only 2.4 doctors per 1000 inhabitants, but in Austria, there are 5.2, and in Lithuania and in Italy: 4.8 (not published). The present data do not indicate the existence of such a dependence, although in most EU countries, vaccinations for children are provided through primary care physicians [31]. For a deeper understanding of the relationship between sociological factors and the use of vaccines, further studies are needed, especially in the EU [32].

### 4.2. Anti-Vaccine Movements

The anti-vaccine movement is not a new phenomenon. As early as the late nineteenth century, there were protests in the UK and the USA against the mandatory smallpox vaccination, arguing that mandatory vaccination was a violation of the right to take care of one’s body in any way one choose [33].

At times, the reason for fear and aversion to vaccinations lay in political decisions; however, the contemporary reason for the lack of trust in vaccination may also be the shortcomings of the procedures concerning controlling scientific integrity, associated with the publication by Andrew Wakefield falsifying the relationship between MMR vaccine and autism as well as enteritis in children. It was when one of the biggest myths regarding vaccination was born, which has been constantly reproduced since its inception in 1998 and fervently cited in discussions, despite the fact that the publication was later withdrawn and debunked repeatedly. The Global Advisory Committee on Vaccine Safety found no link between the vaccine and autism spectrum disorder. The publication of Wakefield’s article caused a long-standing health crisis. The Lancet published an apology and evidence that the study was based on false data. Andrew Wakefield was found guilty of violating professional ethics and removed from the British Medical Register. Even though the myth has long been refuted, distrust of vaccination has begun to spread around the world. Opponents of vaccination, despite the evidence and indisputable facts, continue to use Wakefield’s conclusions in their anti-vaccination campaigns [34].

Anti-vaccination movements make use of modern channels of communication, such as social media, and, since 2019, social media accounts run by Vaccine-Hesitant Men (or anti-vax) are followed by at least 7.8 million more people than before [35]. According to Burki, anti-vaccine activists reach 12% of the total audience following the movement. There is also a dangerous group of entrepreneurs who reach around half of the anti-vaccine movement followers and expose them to advertisements of products with alleged health benefits [35].

Social attitudes towards vaccination can be divided into three categories: there are people who are vaccinated willingly (in the UK and the USA: 70–90% of the population), dogmatic opponents of vaccination who will not change their views, and undecided people who ask valid questions, and it is this group of people that should be focused on, educated, and informed [35].

The anti-vaccine movement is perhaps one of the reasons for the growing incidence of measles. ECDC recognized the seriousness of the problem and prepared a document entitled *Catalogue of interventions addressing vaccine hesitancy,* a practical tool for public health organizations and immunization stakeholders [15].

## 5. Conclusions

It should be considered to standardize the calendars and documentation confirming vaccinations at the EU level, which is especially vital in the time of the COVID-19 pandemic. The electronic version of such a document would allow access to information concerning past vaccinations, both for EU citizens traveling within the EU, as well as for newly arrived immigrants.

Moreover, the situation in Poland (rubella, mumps) and in Lithuania (N. meningitidis) indicates that the response of the epidemiology of VPCD to any changes in vaccination is a long-term one. This is a public health component that takes time to show, and the situation can only improve after many years of intensive work on an effective vaccination schedule. The epidemiological situation of Finland, Sweden, Norway, Denmark, and the Netherlands (IPD) proves the importance of an appropriate selection of serotypes for vaccine production.

## Figures and Tables

**Figure 1 vaccines-09-00796-f001:**
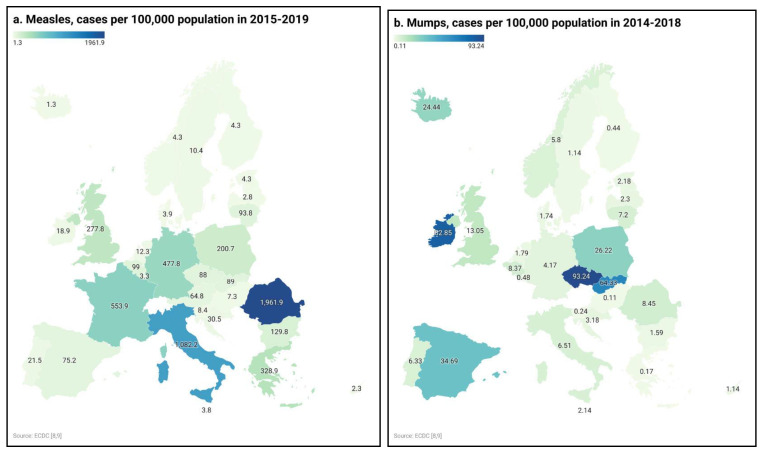
Long-term selected vaccine-preventable childhood diseases incidence rates in the EU and EFTA countries: (**a**) measles (2015–2019), (**b**) mumps (2014–2018), (**c**) rubella (2014–2019), (**d**) *Streptococcus pneumoniae* (2014–2018), (**e**) *Neisseria meningitidis* (2014–2017).

**Table 1 vaccines-09-00796-t001:** Incidence of measles, mumps, and rubella in the EU and EFTA countries in selected years.

Country	Measles					Mumps					Rubella					
	Cases per 100,000 Population					Cases per 100,000 Population					Cases per 100,000 Population					
	2015	2016	2017	2018	2019	2014	2015	2016	2017	2018	2014	2015	2016	2017	2018	2019
Austria	30.9	2.7	9.5	7.7	14.0	NA	NA	NA	NA	NA	0.1	0.0	0.0	0.4	0.0	0.0
Belgium	4.6	7.8	36.7	11.7	38.2	2.0	1.4	1.3	1.6	2.1	NA	NA	NA	NA	NA	NA
Bulgaria	0.0	0.1	16.5	1.3	111.9	0.5	0.3	0.3	0.2	0.4	0.1	0.1	0.0	0.0	0.0	0.0
Croatia	21.9	0.4	0.7	2.3	5.2	0.8	0.8	0.7	0.4	0.5	0.0	0.0	0.0	0.0	0.0	NA
Cyprus	0.0	0.0	0.3	1.5	0.5	0.2	0.2	0.1	0.2	0.3	0.0	0.0	0.0	0.0	0.0	0.0
Czechia	0.9	0.7	14.6	20.7	51.1	6.3	15.1	53.6	13.2	5.0	0.0	0.0	0.0	0.0	0.0	0.0
Denmark	0.9	0.3	0.4	0.8	1.5	0.7	0.3	0.3	0.2	0.3	0.0	0.0	0.0	0.0	0.0	0.0
Estonia	0.4	0.2	0.1	1.0	2.6	0.8	0.2	0.3	0.5	0.5	0.0	0.0	0.0	0.0	0.0	0.0
Finland	0.2	0.4	1.0	1.5	1.2	0.0	0.0	0.1	0.2	0.1	0	0.1	0	0.0	0.0	0.0
France	36.4	7.9	51.8	291.9	165.9	NA	NA	NA	NA	NA	NA	NA	NA	NA	NA	NA
Germany	246.6	32.6	92.9	54.3	51.4	1.0	0.9	0.9	0.8	0.6	0.2	0.1	0.4	0.1	0.0	0.6
Greece	0.1	0.0	96.7	229.3	2.8	0.0	0.0	0.0	0.1	0.0	0.0	0.0	0.0	0.0	0.0	0.2
Hungary	0.0	0.0	3.6	1.4	2.3	0.0	0.1	0.0	0.0	0.0	0.0	0.0	0.0	0.0	0.0	0.2
Iceland	0.0	0.1	0.3	0.0	0.9	0.0	18.7	2.2	2.8	0.8	0.0	0.0	0.0	0.0	0.0	0
Italy	25.6	86.1	539.9	268.6	162.0	1.4	1.1	1.3	1.4	1.3	NA	0.1	1.4	0.1	0.0	2.8
Liechtenstein	NA	NA	NA	NA	NA	NA	NA	NA	NA	NA	NA	NA	NA	NA	NA	NA
Latvia	0.0	0.0	0.0	2.5	0.3	0.6	1.1	0.3	0.2	0.1	0.1	0.0	0.0	0.0	0.2	0.3
Lithuania	5.0	2.2	0.2	3.0	83.4	1.6	1.4	1.9	1.6	0.7	0.0	0.0	0.8	0	0.0	0.0
Luxembourg	0.0	0.0	0.4	0.4	2.5	0.2	0.0	0.0	0.7	0.2	0.0	0.0	0.0	0.0	0.0	4.0
Malta	0.1	0.0	0.0	0.5	3.2	0.6	0.8	0.4	0.4	0.0	0.0	0.0	0.0	0.0	0.0	6.2
Netherlands	0.7	0.6	1.6	2.4	7.0	0.2	0.5	0.4	0.3	0.4	0.0	0.0	0.0	0.0	0.0	0.0
Norway	1.4	0.0	0.0	1.2	1.7	0.3	3.4	1.6	0.3	0.2	0.1	0.0	0.0	0.0	0.0	0.3
Poland	4.8	13.3	6.3	34.0	142.3	6.6	5.8	5.2	4.4	4.2	15.6	5.3	0.3	1.4	1.2	4.0
Portugal	0.0	0.0	3.4	17.1	1.0	0.8	1.4	1.3	1.7	1.0	0.1	0.1	0.0	0.0	0.1	0.2
Rep. Of Ireland	0.2	4.3	2.5	7.7	4.2	14.9	40.6	9.8	5.9	11.7	0.1	0.1	0. 1	0.0	0.1	0.1
Romania	0.7	243.2	907.6	639.8	170.6	0.6	2.3	3.3	1.6	0.6	0.1	0.0	8.2	0.1	0.0	9.0
Slovakia	0.0	0.0	0.6	56.5	31.9	28.6	31.3	3.7	0.5	0.2	0.0	0.0	0.0	0.0	0.0	6.8
Slovenia	1.8	0.1	0.8	0.9	4.8	0.1	0.1	0.0	0.1	0.0	0.0	0.0	0.0	0.0	0.0	1.2
Spain	5.5	3.8	15.7	22.6	27.6	2.0	3.3	5.5	12.3	11.5	0.0	0.0	0.0	0.0	0.0	0.0
Sweden	2.2	0.3	3.0	2.6	2.3	0.2	0.2	0.2	0.3	0.2	0.0	0.0	0.0	0.3	0.0	0.0
United Kingdom	9.2	57.1	28.0	95.3	88.2	4.2	2.6	1.5	3.0	1.7	0.0	0.0	0.0	0.0	0.0	0.0
Median	0.8	0.4	2.8	2.8	5.0	0.7	1.0	0.8	0.4	0.4	0.0	0.0	0.01	0.0	0.0	0.2
1Q	0.025	0	0.4	1.325	2.3	0.2	0.2	0.3	0.2	0.2	0.0	0.0	0.0	0.0	0.0	0.0
3Q	4.8	3.8	15.7	22.6	51.1	1.7	2.8	2.0	1.7	1.1	0.1	0.0	0.0	0.0	0.0	2.0

EU, European Union; EFTA, European Free Trade Association, NA not available; 1Q, first quartile; 3Q third quartile.

**Table 2 vaccines-09-00796-t002:** Incidence of *Streptococcus pneumoniae* and *Neisseria meningitides* in the EU and EFTA countries in selected years.

Country	*Streptococcus Pneumoniae*					*Neisseria Meningitidis*			
	Cases per 100,000 Population					Cases per 100,000 Population			
	2014	2015	2016	2017	2018	2014	2015	2016	2017
Austria	3.8	4.9	5.0	6.2	6.9	0.4	0.3	0.4	0.2
Belgium	10.3	11.8	11.5	12.7	13.5	0.8	0.9	0.9	0.8
Bulgaria	0.3	0.4	0.5	0.5	0.3	0.2	0.1	0.1	0.1
Croatia	0.6	0.6	0.3	0.4	0.5	0.8	1.0	0.7	0.9
Cyprus	1.6	1.1	0.6	2.3	2.0	0.5	0.5	0.7	0.5
Czechia	3.2	3.9	3.1	3.7	5.0	0.4	0.5	0.4	0.6
Denmark	12.9	14.3	12.8	13.4	13.8	0.8	0.4	0.7	0.7
Estonia	0.9	1.8	2.3	3.4	3.3	0.2	0.3	0.2	0.3
Finland	12.9	14.9	14.9	14.9	13.8	0.4	0.4	0.3	0.3
France	6.6	6.9	7.9	8.0	7.7	0.6	0.7	0.8	0.8
Germany	NA	NA	NA	NA	NA	0.3	0.3	0.4	0.3
Greece	0.3	0.5	0.5	0.5	0.4	0.5	0.5	0.5	0.4
Hungary	1.5	1.9	2.3	2.7	3.4	0.3	0.4	0.5	0.4
Iceland	7.4	7.6	5.7	8.0	8.6	0.3	1.2	0.0	0.9
Italy	1.6	2.1	2.5	2.8	2.6	0.3	0.3	0.4	0.3
Latvia	2.5	4.4	3.3	3.8	3.9	0.3	0.5	0.2	0.4
Liechtenstein	NA	NA	NA	NA	NA	NA	NA	NA	NA
Lithuania	0.2	0.9	1.9	2.7	2.3	1.8	1.9	2.4	2.4
Luxembourg	0.2	0	0	0.2	0.2	0.5	0.2	0.2	0
Malta	5.1	2.0	2.4	3.9	6.5	3.0	1.1	1.3	0.4
Netherlands	13.0	15.8	14.9	14.4	16.0	0.5	0.5	0.9	1.2
Norway	11.1	10.1	11.5	10.6	11.0	0.4	0.4	0.5	0.3
Poland	1.9	2.6	2.5	3.1	3.6	0.5	0.6	0.4	0.6
Portugal	NA	1.4	1.6	2.9	3.9	0.5	0.6	0.4	0.5
Rep. Of Ireland	7.4	7.6	8.0	8.7	10.6	1.6	1.5	1.8	1.5
Romania	0.3	0.3	0.3	0.3	0.4	0.3	0.3	0.3	0.3
Slovakia	1.4	1.3	1.1	1.8	1.8	0.4	0.4	0.4	0.7
Slovenia	13.4	16.1	13.6	15.9	13.4	0.4	0.8	0.3	0.4
Spain	5.0	5.5	4.9	6.6	6.3	0.3	0.5	0.6	0.6
Sweden	12.0	13.5	13.7	13.7	13.9	0.5	0.5	0.6	0.5
United Kingdom	6.5	8.9	9.5	9.6	9.9	1.2	1.4	1.3	1.2
Median	3.5	3.9	3.1	3.8	5.0	0.5	0.5	0.5	0.5
1Q	1.3	1.3	1.6	2.7	2.3	0.3	0.4	0.3	0.3
3Q	8.1	8.9	9.5	9.6	10.6	0.6	0.8	0.7	0.8

EU, European Union; EFTA, European Free Trade Association, NA not available; 1Q, first quartile; 3Q third quartile.

## Data Availability

Not applicable.

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
