# Peer review of "Incidence of Vaccine-Preventable Childhood Diseases in the European Union and in the European Free Trade Association Countries"

_vaccines, 2021, doi:10.3390/vaccines9070796_

Round 1

Reviewer 1 Report

The Authors present data from European population on vaccine preventable diseases in regard to the current vaccination guidelines. 

Major comments:

  1. The data in table 1 and 2 are presented in a heterogeneous manner. For Measles they are presented as cases per 100,000 population for others as cases in total. This introduces bias in interpretation and is clouding the judgement. My suggestion is to present the cases more uniformly i.e. as cases per 100,000 because the population of European countries is highly variable. It should be in theory possible to calculate cases per population for each country in each year using population data. 
  2. Data on tuberculosis is not discussed and grounds for omission are not stated. Is tuberculosis not meeting the definition of VPCD?
  3. Data from Lichtenstein are omitted, although the low population of this country should not be grounds for omission. 

Minor issues:

  1. Please use country names uniformly throughout the manuscript
  2. I suggest to very firmly state that Wakefield's study was fraudulent, as the readers may not be aware of this issue. https://www.ncbi.nlm.nih.gov/pmc/articles/PMC3136032/
  3. Minor typos and unclear sentences were marked in the manuscript directly. 

Author Response

DETAILED RESPONSE TO REVIEWERS, STEP-BY-STEP REPLIES TO REVIEWERS' COMMENTS:

Reviewer #1:

The Authors present data from European population on vaccine preventable diseases in regard to the current vaccination guidelines. 

Major comments:

  1. The data in table 1 and 2 are presented in a heterogeneous manner. For Measles they are presented as cases per 100,000 population for others as cases in total. This introduces bias in interpretation and is clouding the judgement. My suggestion is to present the cases more uniformly i.e. as cases per 100,000 because the population of European countries is highly variable. It should be in theory possible to calculate cases per population for each country in each year using population data. 

Authors’ reply: Corrected according to suggestions, .

  1. Data on tuberculosis is not discussed and grounds for omission are not stated. Is tuberculosis not meeting the definition of VPCD?

Authors’ reply: For our analysis, we have chosen a few viral and bacterial acute diseases with high epidemic potential, qualifying acute diseases as developing suddenly and lasting a short time, often only a few days or weeks, and accompanied by distinct symptoms that require urgent or short-term care.

  1. Data from Lichtenstein are omitted, although the low population of this country should not be grounds for omission. 

Authors’ reply: Corrected according to suggestions.

Minor issues:

  1. Please use country names uniformly throughout the manuscript

Authors’ reply: Corrected according to suggestions.

  1. I suggest to very firmly state that Wakefield's study was fraudulent, as the readers may not be aware of this issue. https://www.ncbi.nlm.nih.gov/pmc/articles/PMC3136032/

Authors’ reply: Corrected according to suggestions, as below:

“(…) The publication of Wakefield's article caused a long-standing health crisis. The Lancet published an apology and evidence that the study was based on false data. Andrew Wakefield was found guilty of violating professional ethics and removed from the British Medical Register. Even though the myth has long been refuted, distrust of vaccination has begun to spread around the world. Opponents of vaccination, despite the evidence and indisputable facts, continue to use Wakefield’s conclusions in their anti-vaccination campaigns [34]. (…)”

  1. Minor typos and unclear sentences were marked in the manuscript directly. 

Authors’ reply: Corrected according to suggestions, I am sorry for mistakes!

Reviewer 2 Report

Thank you for the opportunity to review the manuscript on this interesting and important subject.

The text would benefit from a review by a native English speaker. For example, instead of 'obligatory' use 'mandatory'.

- all abbreviations and acronyms should be written out in full at first use, throughout
- Abstract introduction says 'comparing the incidence ... in 2014-2019' but it should say comparing the annual incidence in this period

Main text
Introduction
- while interesting, the first paragraph could be omitted
- the second paragraph should have more references
- countries of EU should be listed somewhere (could change over time)
- page 2, regarding "evaluate whether and how the systemic surveillance of VPCDs contributes to the improvement of the safety of children and newborns in the EU and EFTA countries", I think the question is not addressed in later sections, and possibly too big (as a topic) for this publication
- page 3 point b. Figure 1 shows incidence over five years, but the values are associated with population size. Suggest standardising to for example per 10M citizens. Note population size changes over time too.
- page 4 "differences are not linear" does not make sense to me
- Results section, try to avoid "very low" and "very high" as these are subjective terms
- Table 1 title, make it clear that the numbers are 6-year incidence values (14 15 16 17 18 19)
- Table 1, why are only the values for measles standardised (to 100,000)? Should be for all.
- Table 2, similar issues
- Table 2, check 2 typing errors in the 3 bottom rows
- page 7, it is not very scientific to say something is significantly increased when the term is not defined. I suggest to include in the Methods the definition of significant change. Consult a statistician. Report standardised values (e.g. percentage change). Note the issue of regression to the mean may need to be considered when comparing annual incidences over the years
- page 7, median/mean number of cases is not very useful as it depends on the mix of small/large countries. Report standardised values e.g. per 100,000 capita or similar. Consult an epidemiologist.
- page 7, in paragraph for rubella, values reported as per 100,000 but Table 1 has the same values as cases. Which one is correct?
- page 9, would have been good to see an attempt made to link the country-level vaccination schedule (or the lack of it) to the incidence observed, in the Results section, since it is discussed in the Discussion section
- page 9, what is the difference between 'mandatory', 'compulsory', and 'obligatory'? Use consistent wording if possible
- page 11, instead of 'not published', detail the actual source, e.g. 'personal correspondence'
- page 12, not sure what the first paragraph of 'Conclusions' is based on. I understand there are a lot of very important aspects, but it would be advisable to reduce the scope of this publication and draw conclusions only from the actual findings of the study, and map out the direction of future research

Author Response

DETAILED RESPONSE TO REVIEWERS, STEP-BY-STEP REPLIES TO REVIEWERS' COMMENTS:

Reviewer #2:

Thank you for the opportunity to review the manuscript on this interesting and important subject.

Authors’ reply: Thank you for this comment!

The text would benefit from a review by a native English speaker. For example, instead of 'obligatory' use 'mandatory'.

Authors’ reply: Corrected according to suggestions.

- all abbreviations and acronyms should be written out in full at first use, throughout

Authors’ reply: Corrected according to suggestions.

- Abstract introduction says 'comparing the incidence ... in 2014-2019' but it should say comparing the annual incidence in this period

Authors’ reply: Corrected according to suggestions.

Main text, Introduction

- while interesting, the first paragraph could be omitted, the second paragraph should have more references
Authors’ reply: Corrected according to suggestions, we have deleted the first paragraph and supplemented the second paragraph with references.

- countries of EU should be listed somewhere (could change over time)
Authors’ reply: Corrected according to suggestions, the “Introduction” section has been shortened and supplemented with the list of countries.

- page 2, regarding "evaluate whether and how the systemic surveillance of VPCDs contributes to the improvement of the safety of children and newborns in the EU and EFTA countries", I think the question is not addressed in later sections, and possibly too big (as a topic) for this publication
Authors’ reply: Corrected according to suggestions, as below:

“(…) The objective of the study was to compare the incidence of selected VPCDs in the European Union (EU) and European Free Trade Association (EFTA) countries in 2014–2019 taking into account the existing vaccination programs and an attempt to evaluate whether and how the systemic surveillance of VPCDs contributes to a better public health and public safety in the EU and EFTA countries.”

- page 3 point b. Figure 1 shows incidence over five years, but the values are associated with population size. Suggest standardising to for example per 10M citizens. Note population size changes over time too.
Authors’ reply: Corrected according to suggestions, the “Results” section, table 1 and 2 same as Figure 1 have been corrected.

- page 4 "differences are not linear" does not make sense to me
Authors’ reply: Corrected according to suggestions, as below:

“(…) The incidence associated with the discussed infectious diseases varied greatly with no evident pattern or clear trend. (…)”

- Results section, try to avoid "very low" and "very high" as these are subjective terms
Authors’ reply: Corrected according to suggestions.

- Table 1 title, make it clear that the numbers are 6-year incidence values (14 15 16 17 18 19)
Authors’ reply: Corrected according to suggestions: The years selected for analysis were 2014–2019 (or shorter intervals, depending on data availability), excluding 2020 when the COVID-19 (coronavirus disease - 2019) pandemic broke out. The “Methodology” section has been supplemented.

- Table 1, why are only the values for measles standardised (to 100,000)? Should be for all.
Authors’ reply: Corrected according to suggestions, the “Results” section, table 1 and 2 same as Figure 1 have been corrected.

- Table 2, similar issues
Authors’ reply: Corrected according to suggestions, the “Results” section, table 1 and 2 same as Figure 1 have been corrected.

- Table 2, check 2 typing errors in the 3 bottom rows
Authors’ reply: Corrected according to suggestions.

- page 7, it is not very scientific to say something is significantly increased when the term is not defined. I suggest to include in the Methods the definition of significant change. Consult a statistician. Report standardised values (e.g. percentage change). Note the issue of regression to the mean may need to be considered when comparing annual incidences over the years
Authors’ reply: Corrected according to suggestions, the “Results” section has been corrected and supplemented with statistical analysis.

- page 7, median/mean number of cases is not very useful as it depends on the mix of small/large countries. Report standardised values e.g. per 100,000 capita or similar. Consult an epidemiologist.
Authors’ reply: Corrected according to suggestions, the “Results” section, table 1 and 2 same as Figure 1 have been corrected and supplemented with statistical analysis.

- page 7, in paragraph for rubella, values reported as per 100,000 but Table 1 has the same values as cases. Which one is correct?
Authors’ reply: Corrected according to suggestions incidence rate of all diseases were shown per 100 000 population.

- page 9, would have been good to see an attempt made to link the country-level vaccination schedule (or the lack of it) to the incidence observed, in the Results section, since it is discussed in the Discussion section
Authors’ reply: Corrected according to suggestions, the “Results” section, has been supplemented with statistical analysis.

- page 9, what is the difference between 'mandatory', 'compulsory', and 'obligatory'? Use consistent wording if possible
Authors’ reply: Corrected according to suggestions (we use “mandatory”).

- page 11, instead of 'not published', detail the actual source, e.g. 'personal correspondence'
Authors’ reply: Corrected according to suggestions

- page 12, not sure what the first paragraph of 'Conclusions' is based on. I understand there are a lot of very important aspects, but it would be advisable to reduce the scope of this publication and draw conclusions only from the actual findings of the study, and map out the direction of future research

Authors’ reply: corrected according to suggestions. The scope of the paragraph has been narrowed.

Round 2

Reviewer 1 Report

The authors addressed all the issues I raised.

One additional concern that occured in this revision is the unexpected appearance of an additional co-author - Jakub Marchewka.  

I would like to remind the authors that they should follows the ICMJE guidelines for authorship. Please clarify the role of Dr. Marchewka in the second version of the manuscript and why wasn't he listed as a coauthor in the first version of the manuscript. 

Author Response

The authors addressed all the issues I raised.

Authors’ reply: Thank you for this comment!

One additional concern that occurred in this revision is the unexpected appearance of an additional co-author - Jakub Marchewka. I would like to remind the authors that they should follows the ICMJE guidelines for authorship. Please clarify the role of Dr. Marchewka in the second version of the manuscript and why wasn't he listed as a coauthor in the first version of the manuscript.

Authors’ reply: The other reviewer recommended that we use statistical methods for our analysis. Dr Jakub Marchewka is the statistician we decided to consult for that. His analysis has been included in the manuscript, which is why he is now one of the co-authors.

Reviewer 2 Report

Thank you for responding positively to my recommendations and concerns. Additional items noted today:

 Introduction, first paragraph: could you replace "quite common" with a numerical value/range? 'Common' can be subjective. I think there is a guideline in the context of medication side effects for the use of terms such as 'rare' etc. that you could refer to.

Figure 1. Need to re-think what is presented here. The numbers could be incidence values standardised to 100,000 population. If they are incidence rates they would need to be reported per something like 100,000 person-years. Depends on what the numbers are. Most countries in Europe have declining population sizes. Specify the year when the population sizes used were applicable.

Methods, last paragraph. I am not convinced that the statistical methods used were appropriate. How can regression be linear and logistic at the same time? The dependent variable was the count of infections, but in which year (or a sum over a period)? Instead of 'country-level vaccination schedule' use 'mandatory vs non-mandatory policy' to make it clear that you used hypothesis testing based on a 2x2 table.

Conclusions: note 'especially vital' sounds too dramatic!

Author Response

DETAILED RESPONSE TO REVIEWERS, STEP-BY-STEP REPLIES TOREVIEWERS' COMMENTS:

 Methods, last paragraph. I am not convinced that the statistical methods used were appropriate. How can regression be linear and logistic at the same time?

Authors’ reply: Thank you for your time reviewing our manuscript, valuable feedback and favorable reception . Corrected according to suggestions, as below:

“(…) To compare the mandatory vs non-mandatory policy to the annual incidence observed [cases per 100 000 population / year] we used generalized linear regression model. (…)”

The dependent variable was the count of infections, but in which year (or a sum over a period)?

Authors’ reply: The dependent variable is described in the “Methodology, data sources” section in the following way (the description has been corrected in order to be more transparent):

“(…) “Depending on the disease, the data for different years were available: 2015-2019 for measles, 2014-2018 for mumps, 2014-2019 for rubella, 2014-2018 for Streptococcus pneumoniae, and 2014-2017 for Neisseria meningitidis. For each disease and country, the infection cases from the available years were summed up and then presented as number of cases in the period per 100,000 population”

Therefore, the dependent variable was the number of infections per 100 thousand population over the period, with different periods for each disease.

Instead of 'country-level vaccination schedule' use 'mandatory vs non-mandatory policy' to make it clear that you used hypothesis testing based on a 2x2 table.

Authors’ reply: Corrected according to suggestions